# Does Pyroptosis Play a Role in Inflammasome-Related Disorders?

**DOI:** 10.3390/ijms231810453

**Published:** 2022-09-09

**Authors:** Jiajia Zhang, Stefan Wirtz

**Affiliations:** 1Medizinische Klinik 1, Universitätsklinikum Erlangen, Friedrich-Alexander-Universität Erlangen-Nürnberg, 91052 Erlangen, Germany; 2Medical Immunology Campus Erlangen, Friedrich-Alexander-Universität Erlangen-Nürnberg, 91052 Erlangen, Germany

**Keywords:** inflammasomes, inflammasome disease, RCD, pyroptosis, apoptosis, necroptosis

## Abstract

Inflammasomes are multiprotein complexes orchestrating intracellular recognition of endogenous and exogenous stimuli, cellular homeostasis, and cell death. Upon sensing of certain stimuli, inflammasomes typically activate inflammatory caspases that promote the production and release of the proinflammatory cytokines IL-1β, IL-1α, and IL-18 and induce a type of inflammatory cell death known as “pyroptosis”. Pyroptosis is an important form of regulated cell death executed by gasdermin proteins, which is largely different from apoptosis and necrosis. Recently, several signaling pathways driving pyroptotic cell death, including canonical and noncanonical inflammasome activation, as well as caspase-3-dependent pathways, have been reported. While much evidence exists that pyroptosis is involved in the development of several inflammatory diseases, its contribution to inflammasome-related disorders (IRDs) has not been fully clarified. This article reviews molecular mechanisms leading to pyroptosis, and attempts to provide evidence for its possible role in inflammasome-related disorders, including NLR pyrin domain containing 3 (NLRP3) inflammasome disease, NLR containing a caspase recruitment domain 4 (NLRC4) inflammasome disease, and pyrin inflammasome disease. Although the specific mechanism needs further investigations, these studies have uncovered the role of pyroptosis in inflammasome-related disorders and may open new avenues for future therapeutic interventions.

## 1. Introduction

Innate immunity is the body’s first line of defense against foreign invasion. A crucial feature of innate immunity is the capacity to recognize pathogen-associated molecular patterns (PAMPs) and danger-associated molecular patterns (DAMPs) through pattern recognition receptors (PRRs), which results in the activation of key downstream signaling pathways. In 2002, Martinon et al. initially described the inflammasome as a group of intracellular proteins mediating the activation of potent inflammatory factors [1,2,3,4,5,6]. To date, multiple inflammasomes have been characterized [7], and in general, the family of nucleotide-binding oligomerization domain-like receptors (NLRs) as well as absent in melanoma 2-like receptors (ALRs) are important components of the inflammatory complex. Upon sensing of PAMPs or DAMPs, the associated NLRs or ALRs can oligomerize into caspase-1-containing multiprotein complexes that regulate the activation of inflammatory caspases and promote the expression, maturation, and release of multiple proinflammatory cytokines to trigger inflammatory responses [1,5,6]. Interestingly, in addition to triggering the inflammatory response, activated inflammatory caspases can also induce a type of inflammatory regulated cell death termed “pyroptosis”.

Cell death is a complex cellular process with multiple forms and functions, which is broadly classified as regulated cell death (RCD) or accidental cell death (ACD). Apoptosis, arguably the most widely studied RCD, is mediated by a family of cysteine proteases (caspases) inducing cell shrinkage, chromatin condensation, and cell disintegration. However, the cell membrane remains intact, and apoptosis generally does not trigger inflammation [8,9,10]. By contrast, necroptosis, a cascade of molecular events that are usually downstream of the signaling proteins PRK1 and RIPK3, leads to increased cell membrane permeability and eventually the release of cellular inclusions causing cell death and inflammatory responses [11]. Unlike apoptosis and necroptosis, pyroptosis is a form of regulated cell death leading to the specific release of inflammatory mediators at the same time as cell death, thereby triggering strong inflammatory reactions [12]. As an important part of innate immunity, pyroptosis can play important immune protective roles during infectious diseases. On the other hand, dysregulated activation of pyroptosis can also lead to excessive inflammatory reactions and severe organ damage [13]. Recently, growing evidence has shown that pyroptosis is involved in many diseases, such as infectious diseases, autoimmune diseases, atherosclerosis, and cancer [14,15,16]. The activation of intracellular PRRs, such as NLR pyrin domain containing 3 (NLRP3), NLR containing a caspase recruitment domain 4 (NLRC4), and pyrin inflammasomes, have been considered as the main signals to cause pyroptosis. However, aberrantly activated inflammasomes have been implicated in a variety of diseases ranging from rare monogenic inflammatory syndromes to common metabolic diseases, as well as colorectal cancer, cardiovascular diseases, neurodegenerative diseases, and autoinflammatory diseases [17,18,19,20,21,22,23]. Given the widely described inflammasome dysfunction in a number of disorders, the prominent role of pyroptosis as a disease driving downstream event of inflammasome activation is increasingly recognized. Here, we review the molecular mechanisms underlying the execution of regulated cell death in particular pyroptosis in inflammasome-associated diseases and attempt to provide evidence for its possible role in inflammatory-associated diseases, opening new avenues for future therapeutic interventions.

## 2. Molecular Mechanisms of Regulated Cell Death

Regulated cell death (RCD) is a genetically controlled process, including apoptosis, necroptosis, autophagy, ferroptosis, pyroptosis, and netosis. Autophagy is an intracellular waste degradation pathway characterized by intracellular formation of autophagosomes [24]. Ferroptosis occurs independently of necroptosis, autophagy, and pyroptosis as a regulated form of cell death resulting from iron-dependent lipid peroxidation [25]. Netosis is a form of immune-related RCD induced by various pathogens or pathogenic stimuli leading to the formation of neutrophil extracellular traps (NETs), and interestingly, the pyroptosis-related protein GSDMD may be involved in its regulation [26,27]. Cell apoptosis, necroptosis, and pyroptosis are associated with specific morphological changes and activation mechanisms, and importantly, their signal transduction is also substantially related to each other [28,29,30].

### 2.1. Apoptosis

Apoptosis occurs under physiological and pathological conditions typically without the release of inflammatory mediators. Apoptotic cells shrink constantly, and the nuclear membrane breaks to form nuclear fragments, forming apoptotic bodies by germination and foaming. A large number of proteases of the caspase family, such as caspase-2/-3/-6/-7/-8/-9/-10, are key proteins that mediate apoptosis signals through intrinsic and extrinsic pathways (Figure 1) [31,32].

In the intrinsic apoptosis pathway, mitochondrial membrane integrity is incomplete due to changes in the intracellular environment induced by factors such as endoplasmic reticulum (ER) stress, excessive reactive oxygen species, and apoptotic molecule B-cell lymphoma-2-associated X protein (Bax) activation. The subsequent release of mitochondrial cytochrome C into the cytoplasm, as well as oxidative stress or Ca^2+^ overload, leads to opening of mitochondrial permeability transition pores, mitochondrial rupture, and the formation of apoptotic bodies. This pathway is initiated by the upstream caspase-9, which activates the executioner caspases 3, 6, and 7 [29,32].

By contrast, the extrinsic pathway is characterized by interactions of death ligands with death receptors on the cellular surface. These death ligands are mainly members of the tumor necrosis factor (TNF) family, including the TNF receptor chains TNFR1 and TNFR2, Fas cell surface death receptor (FAS), and TNF-related apoptosis-inducing ligands [33]. After binding to death receptors, FAS-related death domain proteins and caspase-8 are recruited to assemble the death-inducing signal complex, which ultimately causes mitochondrial rupture and cell apoptosis. This pathway is initiated by caspase-8/-10 and further executed by activated caspase-3 [31,34].

### 2.2. Necroptosis

Necroptotic cell death is mainly characterized by cell swelling and rupture. Thereby, the nucleus, mitochondria, and other organelles of necrotic cells become deformed and swollen, the membrane permeability increases, and finally, the plasma membrane ruptures. Subsequently, the cellular contents containing a variety of proinflammatory molecules release into the extracellular space, triggering inflammatory responses. Necroptosis is tightly regulated by a cascade of signaling molecules. During the necroptosis process, phosphorylation of receptor-interacting protein kinase 1/3 (RIPK1/3) results in phosphorylation and activation of mixed lineage kinase domain-like protein (MLKL), which is inserted into the plasma membrane to form pores [35]. Notably, necroptosis has been reported to occur in both canonical and noncanonical ways (Figure 1) [36].

Among the canonical pathways, TNF-mediated signaling pathways have been widely studied, although activation of other receptors such as Toll-like receptor-3 (TLR-3), TLR-4, and interferon receptors can also lead to programmed necrosis [37]. TNF binds to TNF receptor 1, recruiting TNF receptor-associated death domain, TNF receptor-associated factor 2/5, apoptosis inhibitor protein 1/2, and linear ubiquitin chain assembly complex to form complex I, causing downstream RIPK1 ubiquitination. In response to activation of cylindrical tumor protein or deubiquitinating enzyme A20, TNF receptor and RIPK1 get separated and form complex II. Notably, different types of complexes II (Iia, Iib, Iic) can be distinguished according to their composition and protein activity. Complex Iia/b also includes FAS-related death domain protein, caspase-8, and RIPK1 [36]. Caspase-8 further activates caspase-3/-7 to lead to caspase-8-mediated cell apoptosis. Complex Iic consists of RIPK1, RIPK3, and MLKL, and sequential phosphorylation of these three proteins leads to cell membrane destruction [38].

Nonclassical necrosis pathways have been shown to be mediated by, e.g., lipopolysaccharide (LPS), cytomegalovirus, and human herpesvirus. Cytomegalovirus and human herpesvirus induce programmed necrosis by phosphorylation of RIPK3 through the Z-DNA-binding protein 1 (ZBP) receptor, while LPS induces that through Toll-like receptor 3/4 [39,40,41].

### 2.3. Pyroptosis

Interestingly, pyroptosis has some common morphological characteristics of both apoptosis and necroptosis [42]. Like in apoptosis, nuclear shrinkage and fragmentation of nuclear DNA are commonly observed resulting in positive terminal deoxynucleotidyl transferase (TdT) dUTP nick-end labeling (TUNEL) staining [43], although the overall DNA damage in pyroptotic cells is reduced and the overall structure of the nucleus remains rather intact in comparison. In the early stages of pyroptosis, activated caspase-1 cleaves gasdermin D/E (GSDMD/E) at the junction of its N-terminal and C-terminal structural domains, followed by the binding of the active N-terminal structural domain to phosphatidylinositol on the cell membrane, which forms pores of approximately 10–14 nM in size, causing osmotic swelling of the cell. In the late stage of pyroptosis, swollen cells eventually disintegrate, releasing a large number of inflammatory contents that rapidly stimulate inflammatory tissue responses [29]. Recently, Kayagaki et al. revealed that the protein Ninjurin1 (NINJ1) is essential for pyroptosis-related plasma membrane rupture. They found that NINJ1-deficient macrophages exhibited impaired plasma membrane rupture in response to diverse inducers of pyroptotic, necrotic, and apoptotic cell death and were unable to release numerous intracellular proteins including high mobility group box 1 and lactate dehydrogenase [44]. Pyroptosis is triggered by N-terminal fragments generated after activation of gasdermin family proteins, which insert into the membrane through oligomerization and translocation to form holes [45]. Three main types of pyroptosis have been widely reported in many studies, including the caspase-1-dependent canonical inflammasome pathway, the caspase-4/-5/-11-dependent noncanonical pathway, and the caspase-3-dependent pathway (Figure 2) [46].

The activation of caspase-1 is the hallmark of canonical pyroptosis. After NLR and AIM2 being activated, they combine with ASC (apoptosis-associated speck-like protein containing a caspase recruitment domain) and recruit caspase-1 precursor protein [47,48]. Subsequently, the caspase-1 precursor is cleaved into p10 and p20 subunits resulting in the cleavage of IL-1β and IL-18 precursors, whereas GSDMD is processed to the amino-terminal (GSDMD-N) and carboxyl-terminal (GSDMD-C) fragments. Subsequently, GSDMD-N inserts into the cell membrane through oligomerization and translocation to form holes, resulting in extracellular release of IL-1β and IL-18 and cell death [49,50].

In noncanonical pyroptosis, LPS, a cell wall component of Gram-negative bacteria, can directly activate caspase-4/-5/-11 [51,52]. Moreover, caspase-11 also activates the pannexin 1 (Panx1) channel, inducing noncanonical assembly of NLRP3 inflammasomes, promoting caspase-1-mediated IL-1β and IL-18 maturation, and exacerbating inflammatory responses [53,54]. Aglietti et al. revealed that GSDMD p30 is only detected in the membrane-containing fraction of immortalized macrophages after caspase-11 activation by LPS, forming pores that compromise the integrity of the cell membrane to induce pyroptosis [55].

Recent studies have shown that adenosine triphosphate (ATP) can activate caspase-3 in macrophages that inhibit NLRP3 to induce pyroptosis [56]. Chemotherapeutic drugs can also promote pyroptosis through gasdermin-E cleaved by caspase-3 [57]. When the process of necroptosis is blocked, inactive caspase-8 can also participate in the formation of the inflammasome and lead to pyroptosis [58]. Caspase-8 also activates NLRP12 and promotes pyroptosis in collaboration with NLRP3 and NLRC4 [59]. Mechanically, death ligands bind to the death receptor and interact with the death domain of Fas-associated via death domain (FADD) protein, which subsequently activates caspase-8. Caspase-8 cleaves pro-caspase-3 to caspase-3, while activated caspase-3 processes GSDME to amino-terminal (GSDME-N) and carboxyl-terminal (GSDME-C). Similar to GSDMD-N, GSDME-N can insert into the cell membrane through oligomerization and translocation to form holes, resulting in cell pyroptosis. In addition, the permeability of the outer mitochondrial membrane is increased in the context of viral infections or other death stimuli, resulting in the release of cytochrome C and binding to apoptotic protease activating factor-1 (Apaf-1) to form apoptotic bodies, thereby causing the activation of caspase-9 to cleave pro-caspase-3 and triggering pyroptosis. Strikingly, Xu et al. found that persistent mitochondrial permeability changes elicited by bile acids, calcium overload, or adenine nucleotide translocator 1 activators drive assembly of an Apaf-1-caspase-4/-11 pyroptosome triggering GSDME-dependent pyroptosis [60]. Orzalli et al. found that the Bcl-2 family members Mcl-1 and Bcl-xL inhibited mitochondrial functions and act as guard proteins of virus-mediated protein synthesis inhibition and trigger pyroptosis upon inactivation [61]. Additionally, studies by Rogers et al. indicated that caspase-3 cleaves the GSDMD-related protein GSDME after Asp270 to generate a fragment that targets the plasma membrane to induce secondary pyroptosis [62]. A further interesting study demonstrated that in the absence of caspase-1, GSDME-dependent pyroptosis regulated by caspase-8/-3 could selectively release IL-1α, but not IL-1β [63]. Although this research augmented the insight into novel mechanisms of pyroptosis, the exact role of the GSDME pathway during regulated cell death requires further exploration.

### 2.4. The Interconnection between Pyroptosis and Other Forms of Cell Death

Although apoptosis, necroptosis, and pyroptosis were historically delineated and characterized as distinct and independent RCD modalities, increasing evidence clearly suggests that multiple cross-regulatory effects between these major cell death modalities exist. Indeed, these recent findings have led to the development of the description of the term PANoptosis, which describes an inflammatory pathway of RCD with key features of all these cell death forms. The apoptosis execution protein caspase-3 is able to cleave GSDMD’s aspartic acid at position 87, and the resulting short N-terminal fragment was shown to have no pore-forming ability. However, in the absence of GSDMD, caspase-1-mediated apoptosis can be induced by pyroptosis signal stimulation [64,65]. Chemotherapeutic drugs can activate caspase-3 at least in some tumor cells, which in turn activates GSDME, triggering cell pyroptosis [66]. Activated caspase-8 can inhibit RIPK3-mediated necroptosis and promote the occurrence of apoptosis. Conversely, caspase-8 blockade can lead to RIPK1-, RIPK3-, and MLKL-dependent caspase-independent necroptosis [67]. In addition, activated caspase-8 is able to cleave and activate GSDMD and induce pyroptosis [68]. Therefore, in particular, caspase-8 seems to act as a molecular switch, fine-tuning the three death modes of apoptosis, necroptosis, and pyroptosis [69]. Interestingly, Zheng et al. described that caspase-6 participates in all the three pathways by binding RIPK3 and enhancing the interaction between RIPK3 and ZBP1, which is required for innate immunity and ZBP1-NLRP3 inflammasome activation. Subsequently, they confirmed that caspase-6 plays an essential role in host defense against influenza A virus (IAV) infection [70].

In addition, recent studies have also shown some interesting interactions between autophagy and pyroptosis. Saitoh et al. found that deficient autophagy-associated protein ATG16L1-deficiency enhanced IL-1β release and cell lysis following pyroptosis [71]. Moreover, autophagy could engulf and degrade multiple ubiquitin-modified inflammasomes (AIM2, NLRP1, and NLRP3), as well as the inflammasome component ASC to negatively regulate pyroptosis [72,73,74,75]. Autophagy was also reported to prevent pyroptosis by eliminating DAMPs and PAMPs [76], thereby downregulating cleaved GSDMD or inhibiting the caspase-1/GSDMD pathway [77]. While autophagy seems to be also involved in GSDME-mediated pyroptosis, the mechanism is largely unclear. Interestingly, Shi et al. pointed out that autophagy limits the inflammasome activity by direct phagocytosis, while conversely activated inflammasomes could stimulate autophagosome formation [75].

In general, there is multifaceted intermodulation between pyroptosis and other RCDs, which could promote inflammatory disease development and maintain cellular homeostasis through various regulatory mechanisms.

## 3. Pyroptosis and Inflammation

As mentioned above, the caspase-1/-4/-5/-11-mediated pyroptosis pathways have been involved in the release of proinflammatory cytokines through GSDMD-formed pores [78,79,80].

Recent research has uncovered that GSDMD can also promote inflammation through various other mechanisms. On the one hand, GSDMD induces the release of IL-1β in a nonpyroptotic manner to exert pyroptosis-like effects. For example, in neutrophils, after the activation and cleavage of GSDMD, GSDMD-N did not migrate to the membrane to form pores, but moved to azure-phagocytic particles and autophagosomes, releasing IL-1β through the formation of pores in the membrane by autophagy [81]. Activation of caspase-8 in intestinal epithelial cells (IECs) can also activate GSDMD, causing the release of IL-1β-containing vesicles through exocytosis. These inflammatory mediators act as exogenous risk factors to further secrete more IL-1β [82]. On the other hand, besides inhibiting GSDMD-N, GSDMD-C can also combine with the p10 fragment produced by caspase-1/-4 self-processing and promote GSDMD cleavage to further enhance pyroptosis [83,84]. Additionally, pyroptosis, as one type of programmed death, is closely related to other types of cell death. Multiple studies have demonstrated that caspase-3/-8 can be activated in mice with caspase-1 inactivation or deletion, thereby leading to cytolytic death, which may be caused by the inhibition of GSDMD-related pyroptosis [85]. However, the way of cell death downstream is still controversial. Lee et al. suggested that lytic death is cell necrosis secondary to apoptosis [86], whereas Schneider et al. considered that this may be an inflammatory death that is not identical to either apoptosis or pyroptosis [87].

In the late stages of pyroptosis, the cells swell and membranes rupture, releasing large amounts of inflammatory components including mature IL-1β and IL-18. Although these cytokines were previously assumed to be passively released as a result of cell disintegration, there is now accumulating evidence that secretion precedes plasma membrane rupture in pyroptotic cells. IL-1β binds to the IL-1 receptor to enhance the inflammatory response by triggering NF-κB with accelerated synthesis of proinflammatory agents such as cyclooxygenase-2 and IFN-γ [88]. IL-18 activates the p38-MAPK signaling pathway to increase the release of other inflammatory cytokines, including IL-1α, IL-6, and IL-8 [88], which promotes inflammation. Significantly, IL-1β and IL-18 also induce NETosis in neighboring neutrophils, expanding the inflammatory and immune response [89,90]. Furthermore, HMBG1 released by pyroptotic cells, on the one hand, triggers DAMPs to promote inflammatory cytokine production, on the other hand, binding to RAGE causes macrophage pyroptosis [91,92]. Additionally, pyroptotic cells also release large amounts of ATP, which activates the NLRP3 inflammasome, causing proinflammatory cytokines release [93,94].

Collectively, pyroptosis induces cell disintegration and the release of inflammatory cytokines via different mechanisms. Balanced inflammatory responses could activate immune cells and enhance immunity, whereas a sustained activation of pyroptosis-related pathways may promote diseases.

## 4. Pyroptosis and Inflammasome-Related Disorders

By altering the immune response, activated inflammasomes play essential roles in the context of several inflammatory diseases. Human inflammatory diseases have traditionally been named based on pathologic adaptive immune responses involving excessive antibody responses to self and nonpathogenic external antigens. However, numerous inflammatory diseases fail to merge into this classification, including common diseases associated with obvious tissue inflammation, such as Inflammatory bowel disease (IBD), gout, and Systemic lupus erythematosus (SLE), and also a number of rare genetic disorders associated with systemic and tissue inflammation, namely hereditary febrile diseases. Because experimental and clinical data clearly suggest increased inflammasome activity, several disorders are collectively referred to as inflammasome-related disorders (IRDs), including NLRP3 inflammasome disease, NLRC4 inflammasome disease, pyrin inflammasome disease, and multifactorial inflammasome diseases (Figure 3).

Seemingly, a very common pathogenic mechanism of inflammasomal diseases is a decreased threshold or abnormal continuous activation of inflammasomes caused by gene mutation. Therefore, pyroptosis shares an upstream signaling pathway with IRDs, which may play important roles in the pathogenesis of IRDs. Accordingly, in some IRD animal models, inflammation is alleviated by suppression of pyroptosis by genetic GSDMD deficiency [95,96]. Moreover, high levels of IL-1β and IL-18 in IRDs patients may be released through the pores in the membrane formed by pyroptosis and rupture of the membrane, although IL-1β maturation in GSDMD-deficient mice is unaffected [49,97]. Importantly, inhibition of IL-1β and IL-18 in inflammasome disease does not completely alleviate inflammation, suggesting that cell death can cause inflammation, which is related to caspase-1 and is likely to be pyroptosis [98].

At present, the relationship between pyroptosis and IRDs has not been fully confirmed as the mechanisms of activation of inflammasomes and subsequent pyroptosis are seemingly different in various etiologies. For example, pyroptosis is induced by bacterial infection, which causes the activation of NLRP3 inflammasomes through a two-step reaction of initiation and activation. However, in IRDs, pyroptosis is triggered by the activation of inflammasomes caused by gene mutation, which usually requires only one step of initiation [99]. Additionally, the activation of the pyrin inflammasomes by bacterial infection requires complete microtubule structures, while the activation of pyrin inflammasome by Mediterranean fever (MEFV) gene mutation does not require microtubule structures [100,101]. Furthermore, IL-1β can be released by GSDMD in a manner unrelated to pyroptosis, which is similar to pyroptosis [82]. Therefore, by studying the role of pyroptosis by knocking out GSDMD in IRDs animal models, it may be impossible to determine whether the experimental results are caused by the inhibition of pyroptosis or the inhibition of the release of IL-1β unrelated to pyroptosis. Thus, the specific relationship between IRDs and pyroptosis still needs further study.

### 4.1. NLRP3 Inflammasome Disease

The NLRP3 inflammasome can be activated either in a canonical or noncanonical manner. On the one hand, it can be activated via caspase-1 by infectious and endogenous ligands such as pore-forming toxins, ATP, and uric acid crystals [102,103,104,105], while on the other hand, caspase-4/-5/-11-mediated LPS sensing also triggers the inflammatory reaction [51,106,107,108]. IRDs related to the NLRP3 inflammasome include both monogenic diseases, such as cryopyrin-associated periodic syndrome (CAPS), and polygenic diseases, such as Crohn’s disease (CD) and gout. This part mainly introduces the relationship between CAPS and pyroptosis (Table 1).

CAPS is triggered by the continuous activation of NLRP3 inflammasomes due to functionally acquired mutations of the NLRP3 gene, causing persistent caspase-1 activation and disproportionate production of IL-1β and IL-18 [109,110]. According to the different symptoms, CAPS can be classified into three types: neonatal-onset multisystem inflammatory disease (NOMID), Muckle–Wells syndrome (MWS), and familial cold autoinflammatory syndrome (FCAS). The common clinical symptoms of these diseases include fever, urticaria, and central nervous system inflammation [111,112].

Unlike MWS, IL-1 antagonists are only partially effective in NOMID and FCAS patients, suggesting that factors other than IL-1β are involved in the pathogenesis of CAPS [98]. Brydges et al. found that nonapoptotic cell death related to caspase-1 could cause inflammation in the FCAS mouse model, which was speculated to be pyroptosis [98]. Furthermore, studies have shown that protein kinase A (PKA) directly phosphorylates a specific site of NLRP3 to inactivate it and inhibit pyroptosis, and some patients with NOMID have abnormal NLRP3 activation and pyroptosis due to mutations at this site [113]. In an in vivo experiment in NOMID using a mouse model on a GSDMD-deficient background, the results showed that symptoms including skin lesions, splenomegaly, and growth restriction were alleviated, and neutrophil infiltrations in the liver, subcutaneous tissue, and spleen were reduced. It was further confirmed that GSDMD-mediated cell pyroptosis played an important role in the pathogenesis of NOMID, and GSDMD was expected to be a new target for NOMID treatment [96].

### 4.2. NLRC4 Inflammasome Disease

NLRC4-related inflammasome diseases mainly include autoinflammation with infantile enterocolitis (AIFEC), NOMID, and FCAS4, while only AIFEC has been reported to be associated with pyroptosis [114,115]. AIFEC, a newly discovered IRD in 2014, is caused by abnormal activation of the NLRC4 inflammasome due to a functionally acquired mutation in the helical domain 1 domain region of the NLRC4 gene. AIFEC mainly manifests as periodic fever, secretory diarrhea, neonatal colitis, and macrophage activation syndrome. Notably, increased pyroptosis can be detected in peripheral blood of AIFEC patients [116,117]. In 2018, Moghaddas et al. found that mutations in the leucine enrichment domain (LRR) of the NLRC4 gene can also cause symptoms similar to AIFEC, but the process of cell pyroptosis caused by LRR mutations does not depend on the involvement of apoptosis proteins (ASC) with lower cytokine response. That means pyroptosis induced by mutations at this site may be downstream of NLRC4 inflammasome and caspase-1 independently of ASC (Table 1) [118].

### 4.3. Pyrin Inflammasome Disease

Pyrin-associated autoinflammatory diseases (PAADs) are a group of IRDs caused by the over-activation of the pyrin inflammasome by MEFV gene mutation, which leads to a series of symptoms, such as familial Mediterranean fever (FMF), pyrin-associated autoinflammation with neutrophilic dermatosis (PAAND), chronic aseptic osteomyelitis, and ulcerative dermatitis. Currently, both FMF and PAAND have been reported to be associated with pyroptosis (Table 1) [119].

FMF encompasses a group of autosomal recessive disorders, and the main clinical manifestations include periodic fever, rash, serositis, and arthritis [120]. The primary cause of FMF is the inactivation of Rho GTPases caused by MEFV gene mutations, which further cause the reduction of the activation threshold of pyrin inflammasomes [101]. Evidence exists that the number of MEFV allele mutations was positively correlated with severity of FMF symptoms and pyroptosis in peripheral blood. Moreover, inhibition of PKN1 and PKN2 proteins (serine/threonine-protein kinase N1/2), which are necessary for pyrin inflammasome activation in peripheral blood of FMF patients profoundly reduced pyroptosis [101]. In recent years, Kanneganti et al. demonstrated that the infection of macrophages in the FMF model with bacteria of the genus Clostridium could cause pyroptosis, accompanied by increased IL-1β secretion. Further in vivo experiments showed that IL-1β levels decreased significantly after GSDMD gene knockout, inflammation was alleviated, and organ-specific inflammatory injuries, such as hepatitis, glomerulonephritis, and colitis were also alleviated [121]. These studies suggest that GSDMD-mediated pyroptosis may play an important role in the pathogenesis of FMF, and GSDMD is expected to be a new target for the treatment of FMF.

PAAND is an autosomal dominant disease with the main clinical manifestations in childhood, which include recurrent neutrophilic dermatitis, periodic fever, joint pain, myalgia, or myositis [122]. Unlike FMF, the pathogenesis of PAAND is caused by mutations in exon 2 of the MEFV gene leading to the disruption of the normal inhibitory state of the pyrin inflammasome and its continued activation, resulting in the massive release of IL-1β and IL-18 and GSDMD-mediated pyroptosis [123]. In patients with PAAND, an increase in pyroptosis can be detected in peripheral blood mononuclear cells [100,122,124,125]. Pyroptosis-mediated membrane pore formation and intracellular DAMP release can cause the production and aggregation of a large number of cytokines in skin and other tissues, which will ultimately lead to neutrophilic dermatitis and inflammation in PAAND patients [125].

### 4.4. Multifactorial IRDs

The pathogenesis of multifactorial IRDs is seemingly the result of a combination of many elements, such as dietary, environmental, genetic, and immune factors. Thereby, studies indicate that pyroptosis downstream of inflammasome activation may also be related to the pathogenesis of CD, gout and SLE and thus may represent a potential therapeutic target in these rather common diseases (Table 1).

CD, with abdominal pain, diarrhea, and other gastrointestinal symptoms as the main clinical manifestations, is an inflammatory bowel disease (IBD) with a complex etiology that could be attributed to a combination of genetic and environmental variables. In some patients with CD, the disease is related to alterations in the NOD2 gene [126]. Previous studies have found that the expression of p20 fragments of caspase-1, NLRP3, and GSDMD in IEC and macrophages are increased in patients with CD, suggesting that pyroptosis of both cell types may be closely related to the loss of intestinal mucosal barrier function [127,128]. Although the epithelium can eliminate some of the pathogen-infected cells by pyroptosis, this process can support barrier disruption and subsequent inflammation [129]. A multicenter study with a cohort of 100 patients with CD found that the severity of pyroptosis in small intestinal IEC can serve as a potential biomarker for disease severity and predict the therapeutic efficacy of the integrin antagonist vedolizumab [130]. NIMA-related kinase 7 (NEK7) is a necessary enzyme for the activation of the NLRP3 inflammasome [131]. Chen et al. found that activation of NLRP3 in IEC can interact with NEK7 to promote the occurrence of pyroptosis in IEC, which plays a crucial role in the pathogenesis of CD. Knockdown of the NEK7 gene in mice with colitis reduced the expression of pyroptosis-related proteins and intestinal inflammatory symptoms, as well as systemic inflammation [127]. Pyroptosis also plays a proinflammatory role in macrophages associated with CD pathogenesis. A recent study showed that macrophages release nucleoprotein spliceosome-associated protein 130 (SAP130) after intestinal mucosal damage in patients with CD and activate NLRP3/caspase-1, thereby promoting GSDMD cleavage and pyroptosis [128]. However, other studies have shown that although there is less intracellular GSDMD in macrophages compared with IEC, and the pyroptosis mediated by GSDMD mainly plays a proinflammatory role, GSDMD itself can also play a protective role in intestinal inflammation. In a murine colitis model, GSDMD alleviates intestinal inflammatory symptoms by inhibiting the cyclic guanosine phospho-adenosine synthase-interferon gene stimulating factor (cGAS-STING) signaling pathway. Notably, this process was not related to the intestinal flora, and was mainly caused by the excretion of large amounts of potassium ions through pore formation in the membrane [132]. In conclusion, the intracellular GSDMD of macrophages may have both proinflammatory and anti-inflammatory effects, although it seems to have mainly proinflammation in CD.

Gout is a multifactorial IRD caused by the deposition of monosodium urate (MSU) crystals in joints and the surrounding tissues, resulting in systemic inflammation [133]. Although previous studies have shown that MSU-induced inflammation is closely related to the activation of the NLRP3 inflammasome [105,134], the role of pyroptosis in the pathogenesis of gout remains controversial [135]. In vitro experiments showed that MSU crystals rapidly increased the expression of GSDMD in mouse macrophages, providing evidence for the excessive occurrence of pyroptosis in gout patients. However, further in vivo experiments showed that MSU crystals did not significantly change cell death or decrease IL-1β levels in mice with GSDMD, caspase-1, or MLKL deficiency. Moreover, inhibition of NLRP3 inflammasome by extracellular high potassium did not significantly affect the occurrence of cell death, suggesting that MSU-induced cell death is not mediated by pyroptosis or necroptosis [136]. However, Li et al. suggested that pyroptosis exerts an important role in the development of gout. Purine guanosine monophosphate receptor P2Y14R negatively regulates NLRP3 inflammasomes, and in mice with knockout of the P2Y14R gene, NLRP3 expression and MSU-related pyroptosis were downregulated, while gout symptoms were alleviated [137]. In conclusion, the role of pyroptosis in the pathogenesis of gout is still controversial, and whether GSDMD can be a target for the treatment of gout needs further investigation.

Systemic lupus erythematosus (SLE) is a complex autoimmune disease involving multiple organs with a highly heterogeneous profile of excessive inflammatory response and tissue damage [138]. Recently, increasing attention has been paid to the relationship between pyroptosis and SLE, although the phenomenon of pyroptosis in SLE remains incompletely understood [139]. Initially, no evidence was available for the presence of GSDMD/GSMDE in SLE patients, but Cao et al. showed significantly increased GSDMD expression and cleavage in kidney tissue from lupus nephritis patients and mice. The combination of mycophenolate mofetil, tacrolimus, and steroids alleviated disease progression in humans and MRL/lpr mice by inhibiting NLRP3/caspase-1/GSDMD-mediated pyroptosis [140]. Subsequently, it was demonstrated that the renal tubules of pristane-induced lupus mice and SLE patients could also express high levels of GSDME. In line, GSDME knockout reduced pyroptosis [141] with effective amelioration of lupus-like features [142]. Moreover, NLRP3 was found to be overactivated in patients with SLE and lupus nephritis [143,144], since, anti-dsDNA antibodies, a hallmark feature of SLE, can activate the NLRP3 inflammasome in monocytes/macrophages of SLE patients by inducing mitochondrial ROS production and activation of the TLR4-NF-κB signaling pathway [145]. Recent evidence also indicates that the NLRP3 inflammasome is significantly upregulated in both bone marrow mesenchymal stem cells and monocytes/macrophages in SLE patients [146,147]. Studies revealed that the selective antagonist of the P2X7 receptor, Leucomalin G, could reduce the severity of nephritis in MRL/lpr mice by inhibiting the activation of NLRP3 inflammation and reducing the release of proinflammatory cytokines [148,149]. Notably, in addition to releasing various inflammatory cytokines to enhance the inflammatory response, cell disintegration during pyroptosis also releases condensed and intact DNA representing a source for antinuclear antibody production [150]. Additionally, reducing the activation of the AIM2 inflammasome, which senses double-stranded nucleic acids in the cytoplasm, inhibits macrophage activation and inflammatory responses, which significantly relieves lupus-like symptoms [151].

## 5. Treatment of IRDs

Abnormal inflammasome activation is believed to be the primary cause of the pathological process of IRDs (Table 1). Nonsteroidal anti-inflammatory drugs (NSAIDs) and corticosteroids are the most common drugs for treating CAPS; however, withdrawal of corticosteroids leads to frequent attack relapses or continuous symptoms [152]. In addition, inhibition of IL-1 signaling has been remarkably beneficial in CAPS patients [153]. Currently, IL-1 antagonists applied in these diseases include canakinumab, rilonacept, and anakinra. Canakinumab is a monoclonal antibody that neutralizes IL-1β in the circulation rather than IL-1α [154,155]. Rilonacept and anakinra are IL-1 receptor antagonists, which prevents IL-1α and IL-1β from binding to the IL-1 receptor and exerting their biological functions [156,157]. Unfortunately, inhibition of IL-1 signaling cannot resolve all symptoms of CAPS patients. However, blockage of NLRP3 or caspase-1 may contribute to the therapy for CAPS patients. It was shown that an inhibitor of caspase-1 dramatically prevented IL-1β secretion from LPS-stimulated peripheral blood mononuclear cells of FCAS patients in vitro [158]. There is further evidence that NLRC4-associated hyperinflammation can be successfully treated by IL-18 inhibition [116,159]. Colchicine has been the main therapeutic drug for FMF by suppressing pyrin oligomerization and interfering with neutrophil migration and adhesion to exert its anti-inflammatory effect. However, only two-thirds of the patients responded positively [160]. Recently, other drugs, such as anakinra, canakinumab, and rilonacept are used for the treatment of FMF and showed some benefit in colchicine nonresponders [161]. Anakinra and adalimumab have been used for the treatment of PAAND patients, even though the effect is weak [162,163]. A further study found that the severity of pyroptosis in small intestinal IEC can serve as a potential biomarker for disease severity and predict the efficacy of the anti-integrin drug Vedolizumab in CD treatment [130]. In addition, ustekinumab and infliximab were also used in treating patients with CD and showed similar efficacy in patients with CD [164,165]. As regards gout, the regularly used medicines to counter gout include NSAIDs, colchicine, adrenocorticotropin hormone, xanthine oxidase inhibitors, and uricosuric drugs [166]. Additionally, amitriptyline is applied in gout therapy by blocking innate immune responses mediated by TLR4 and the IL-1 receptor [167]. Other evidence showed that colchicine and corticosteroids may be effective treatments for acute gout [168,169]. For SLE, several specific inhibitors or bioactive agents, such as baicalein, oleuropein, melatonin, and piperine, attenuate LN development by suppressing NLRP3 inflammasome activation, which may also be involved in other systems of SLE treatment [170,171,172,173]. As mentioned above, Cao et al. pointed out that the combination of mycophenolate, tacrolimus, and steroids inhibited the activation of NLRP3 and caspase-1, which reduced GSDMD cleavage and blocked the pyroptosis to dampen SLE development [140].
ijms-23-10453-t001_Table 1Table 1Potential roles of pyroptosis in the pathogenesis of IRDs.IRDsRelatedInflammasomeClinical ManifestationPotential Roles of PyroptosisTreatmentFCASNLRP3Fever, urticaria, arthralgia, conjunctivitisCaspase-1-related nonapoptotic mode of death was observed in the mouse modelIL-1 inhibitionNLRP3/caspase-1 blockageNSAIDs/corticosteroidsNOMIDNLRP3Fever, urticarial rash, CNS inflammation, sensorineural hearing loss, arthritis, bone overgrowth, intellectual retardationPyroptosis was observed in patients’ peripheral blood; Clinical symptom was alleviated after knocking out GSDMD gene in mouse modelIL-1 inhibitionNLRP3/caspase-1 blockageNSAIDs/corticosteroidsAIFECNLRC4Fever, secretory diarrhea, macrophage activation syndromePyroptosis was observed in patients’ peripheral bloodIL-18 inhibitionFMFPyrinFever, rash, arthritis, serositis (chest pain, abdominal pain), amyloidosisClinical symptom was alleviated after knocking out GSDMD gene in mouse modelIL-1 inhibitionColchicinePAANDPyrinNeutrophilic dermatitis, periodic fever, arthralgia, myalgiaPyroptosis was observed in patients’ peripheral bloodAnakinraAdalimumabCDNLRP3Abdominal pain, diarrhea, abdominal mass, weight loss, intestinal obstructionEliminated IEC infected by pathogens; destroyed the intestinal mucosal barrier; promoted intestinal inflammation by activating NLRP3/caspase-1 through AC-cAMP-PKA or Mincle-Syk signaling pathwayVedolizumabUstekinumabInfliximabGoutNLRP3Arthritis, fever, kidney damageThe pyroptosis level was decreased by inhibiting the activation of MSU-associated NLRP3 inflammasomesNSAIDs, amitriptyline, Colchicine, corticosteroids adrenocorticotropin hormone, xanthine oxidase inhibitors, uricosuricSLENLRP3Arthritis, fever, chest pain, mouth ulcers, swollen lymph glands, kidney damage, and facial rashes.Disease progression was alleviated by inhibiting NLRP3/caspase-1/GSDMD-mediated pyroptosis in humans and MRL/lpr miceNLRP3/caspase-1 blockageGSDMD/GSDME inhibitionbaicalein, oleuropein, melatonin, and piperineIRDs, inflammasome-related disorders; FCAS, familial cold autoinflammatory syndrome; NSAIDs, nonsteroidal anti-inflammatory drugs; NOMID, neonatal-onset multisystem inflammatory disease; CNS, central nervous system; AIFEC, autoinflammation with infantile enterocolitis; FMF, familial Mediterranean fever; PAAND, pyrin-associated autoinflammation with neutrophilic dermatosis; CD, Crohn’s disease; IEC, intestinal epithelial cell; MSU, monosodium urate; SLE, Systemic lupus erythematosus.


## 6. Conclusions

Pyroptosis is an important form of RCT characterized by pore formation by gasdermin proteins, cellular lysis and the release of inflammatory cytokines. Inflammatory mediators released during pyroptosis recruit and activate immune cells, which enhance immunity to help the host eliminate pathogens. However, unbalanced activation of pyroptosis may lead to a sustained inflammatory response disrupting host homeostasis and thereby inflammasome activation and its downstream events such as aberrant cytokine release and pyroptosis has emerged as a critical mechanism in various inflammatory diseases. There is a close relationship between IRDs and pyroptosis, and they interact and influence each other. Although pyroptosis has been observed in the peripheral blood of patients with IRDs, and a large number of experiments have demonstrated that IRDs and cellular pyroptosis share the same upstream signaling pathways and modulators, the direct role of gasdermin activation and in IRDs is still incompletely understood. Therefore, investigating further the precise mechanism of pyroptosis execution, developing novel activators or inhibitors of key molecules involved in inflammasome activation, as well as pyroptosis, and combining them with other immunotherapies could potentially pave the way for the discovery of new therapeutic intereventions for IRDs. Since, pyroptosis could be the upstream source or downstream result of other immune or inflammatory responses, we should not only consider the pathogenic effect of pyroptosis, but instead should judge its overall effects in the context of the whole inflammatory or immune cascade response.

## Figures and Tables

**Figure 1 ijms-23-10453-f001:**
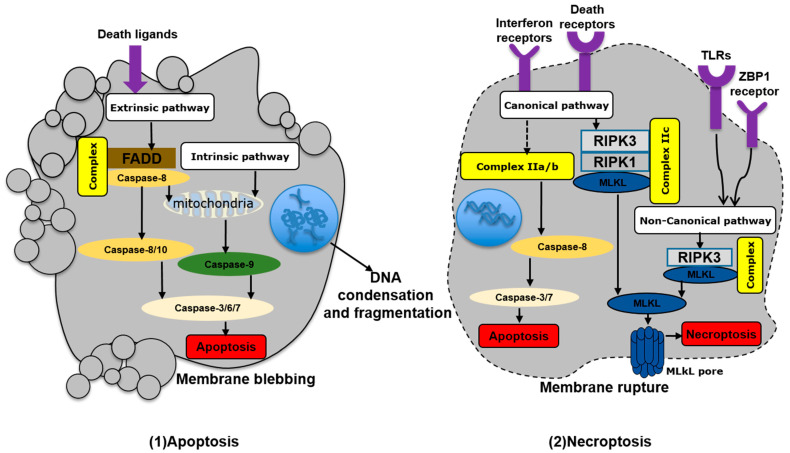
Simplified model of apoptosis and necroptosis. FADD, FAS-associated death domain; RIPK1/3, receptor-interacting protein kinase 1/3; MLKL, mixed lineage kinase domain-like; TLRs, Toll-like receptors; ZBP1, Z-DNA-binding protein. Arrows indicate activation or facilitation.

**Figure 2 ijms-23-10453-f002:**
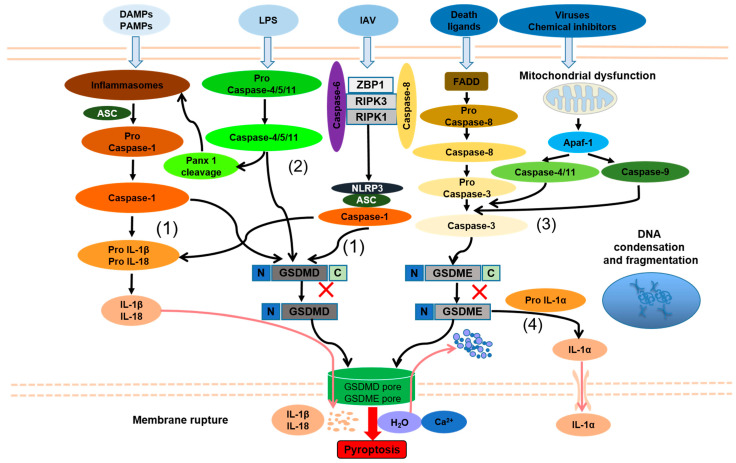
Molecular features of pyroptosis. (1) Caspase-1-dependent pathway of pyroptosis; (2) caspase-4/-5/-11-dependent pathway of pyroptosis; (3) caspase-3-dependent pathway of pyroptosis; (4) caspase-1-independent pyroptosis selectively release IL-1α. PAMPs, pathogen-associated molecular patterns; DAMPs, danger-associated molecular patterns; ASC, apoptosis-associated speck-like protein containing a caspase recruitment domain; IL-1β, interleukin-1β; IL-18, interleukin-18; LPS, lipopolysaccharide; Panx1, Pannexin 1; GSDMD-N/C, N/C-terminal of gasdermin D; GSDME-N/C, N/C-terminal of gasdermin E; IAV, influenza A virus; ZBP, Z-DNA-binding protein 1; RIPK1/3, receptor-interacting protein kinase 1/3; NLRP3, NLR pyrin domain containing 3; FADD, FAS-associated death domain; Apaf-1, apoptotic protease activating factor-1; IL-1α, interleukin-1α. Red cross means the activated caspases cleave gasdermin D/E (GSDMD/E) at the junction of its N/C-terminal structural domains to produce gasdermin D/E N-terminal (NT) fragments; Arrows indicate activation or facilitation.

**Figure 3 ijms-23-10453-f003:**
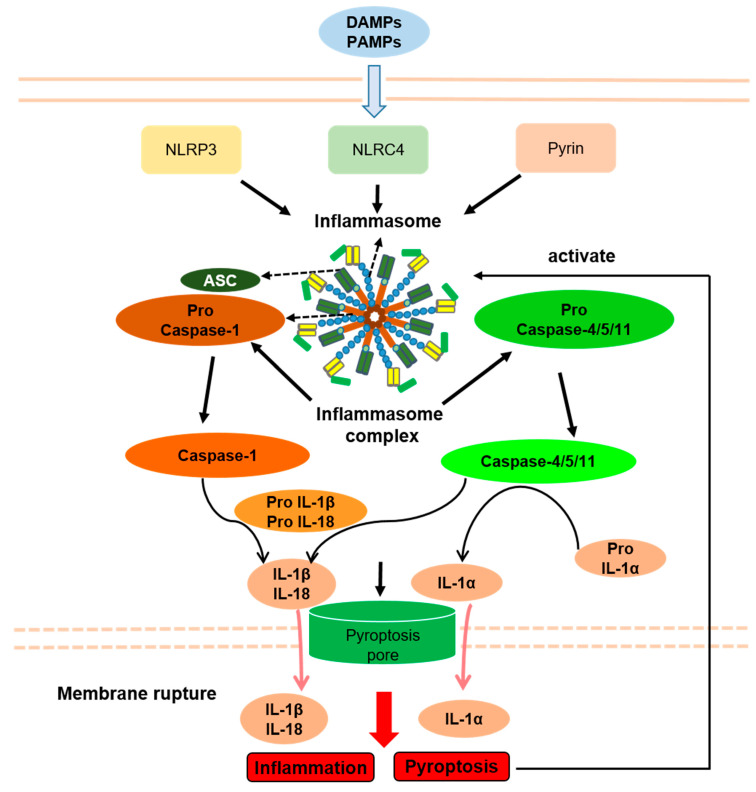
Inflammasomes and Inflammasome-related disorders association with pyroptosis. PAMPs, pathogen-associated molecular patterns; DAMPs, danger-associated molecular patterns; NLRP3, NLR pyrin domain containing 3; NLRC4, NLR containing a caspase recruitment domain 4; ASC, apoptosis-associated speck-like protein containing a caspase recruitment domain; IL-1β, interleukin-1β; IL-18, interleukin-18; IL-1α, interleukin-1α. Arrows indicate activation or facilitation.

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
