# Peer review of "Does Pyroptosis Play a Role in Inflammasome-Related Disorders?"

_ijms, 2022, doi:10.3390/ijms231810453_

Round 1
Reviewer 1 Report
In manuscript ijms-1822218, the authors aimed at describing and discussing the current views on the molecular mechanisms leading to pyroptosis and its putative role in inflammasome-related disorders.
The topics is of great interest. Nonetheless, improvements for some sections could be achieved by better focusing on these questions and by providing a more related background (listed as follows in the specific points).
Specific points:
1. The authors should include more details on the molecular regulation, activation and composition of inflammasome complexes, since this background is required to understand the topic of this review. This can be included, at least, in schemas, table and/or text box.
2. As opposed, the description of the other types of cell death is too much developed here. This is per se out of the focus of this review, especially because: i/ each cell death stands byn its own as very complex pathway and ii/ the aspects in other cell deaths the more related to pyroptosis are indeed the Caspases. As schemas are already included, a very short section and/or text box be should be sufficient. In addition, it would be more appropriate to present first the pyroptosis, and then inclusion as brief description of other types of cell death and focusing on the intersect and comparison to pyroptosis.
3. It would be of interest to better describe the release of inflammatory components upon pyroptosis, especially as it seems to be a primarily aspect in the inflammatory diseases.
Minor points:
1. As this is an active area of research, the authors should also include the recent reports published in 2022.
2. Conclusion and/or consensus view are missing for some paragraphs.
3. The authors should reduce the number of abbreviations in the text by including them only when fully necessary and used to a large extend.
Reviewer 2 Report
The manuscript reviewed the molecular mechanisms leading to pyroptosis and provided the evidence for the possible role of pyroptosis in inflammasome-related disorders (IRDs). The article has summarized a broad range of representative examples of IRDs and the possible underkying mechanisms. Below are comments for the manuscript:
1. L142-145: Certain typical features of pyroptosis are missing, including cell-blebbing and DNA damage with positive staining of TUNEL, these should be described in the review.
2. L252-253: Although pyroptosis shares an upstream signaling pathway with IRDs and plays important roles in the pathogenesis of IRDs, the possible involvement of autophagy cannot be neglected as there is cross-talk between pyroptosis and autophagy.
3. L340-384: Given the various examples discussed, the potential role of pyroptosis in systemic lupus erythematosus should also be discussed as recent studies demonstrated the trigger of NLPR3 inflammasomes in monocytes and bone marrow-derived mesenchymal stem cells.
4. A summary or conclusion is needed for the review to summarize the key messages and the future outlook of this topic.
5. Others:
L95: “Fas Cell Surface Death Receptor” is not a ligand.
L141: Define the abbreviations NINJ1, HMGB1, LDH, MEFV, …etc.
L283: “Muckle-Wells”
Reviewer 3 Report
I have gone through the manuscript. The topic is indeed but authors did not provide a mechanistic overview of the diseases. Different pathologies like cancers or any other disease should be explained comprehensively as an example for a detailed analysis of signaling cascade.
Authors should initially explain the different between pyroptosis and apoptosis.
Animal model studies are not included. Therefore, inclusion of different aspects in animal models will prove to be helpful.
Molecular hierarchy of signaling molecules in necessary for the activation of pyroptosis. How strong is the evidence related to the involvement of transduction cascades in the regulation of pyroptosis?
Round 2
Reviewer 3 Report
Looks in acceptable form now.